# Autophagy is a critical regulator of memory CD8+ T cell formation

Daniel J Puleston[1]*, Hanlin Zhang[1†], Timothy J Powell[1†], Elina Lipina[1], Stuart Sims[2], Isabel Panse[1], Alexander S Watson[1], Vincenzo Cerundolo[1], Alain RM Townsend[1‡], Paul Klenerman[2‡], Anna Katharina Simon[1]*

[1]MRC Human Immunology Unit, Weatherall Institute of Molecular Medicine, University of Oxford, John Radcliffe Hospital, Oxford, United Kingdom; [2]Peter Medawar Building for Pathogen Research, University of Oxford, Oxford, United Kingdom

**Abstract** During infection, CD8+ T cells initially expand then contract, leaving a small memory pool providing long lasting immunity. While it has been described that CD8+ T cell memory formation becomes defective in old age, the cellular mechanism is largely unknown. Autophagy is a major cellular lysosomal degradation pathway of bulk material, and levels are known to fall with age. In this study, we describe a novel role for autophagy in CD8+ T cell memory formation. Mice lacking the autophagy gene Atg7 in T cells failed to establish CD8+ T cell memory to influenza and MCMV infection. Interestingly, autophagy levels were diminished in CD8+ T cells from aged mice. We could rejuvenate CD8+ T cell responses in elderly mice in an autophagy dependent manner using the compound spermidine. This study reveals a cell intrinsic explanation for poor CD8+ T cell memory in the elderly and potentially offers novel immune modulators to improve aged immunity.

**\*For correspondence:** daniel. puleston@ndm.ox.ac.uk (DJP); katja.simon@ndm.ox.ac.uk (AKS)

[†]These authors contributed equally to this work

[‡]These authors also contributed equally to this work

**Competing interests:** The authors declare that no competing interests exist.

**Reviewing editor**: Beth Levine, UT Southwestern Medical School, United States

## Introduction

Upon successful clearance of a pathogen, the majority of short-lived effector T cells die and the remaining cells differentiate into a memory T cell population that provides long lasting immunity. While the cytokines, surface molecules, and signaling components involved in T cell memory formation have been extensively studied, the molecular pathways supporting these cell fate decisions are poorly understood. The T memory population is (1) quiescent, (2) long-lived and actively maintained, and (3) relies on mitochondrial respiration (*Pearce and Pearce, 2013*). Both reduced cell cycling and longevity of this stem cell-like population demand rigorous maintenance of the cytoplasm as debris cannot be diluted to daughter cells, reminiscent of true stem cells (*Mortensen et al., 2011*; *Guan et al., 2013*; *Warr et al., 2013*). Across mammalian cell types, clearance of debris and damaged organelles such as mitochondria is typically executed via autophagy (*Choi et al., 2013*).

A recent study showed that formation of the influenza-specific B cell memory pool requires autophagy (*Chen et al., 2014*). However, while the role of autophagy in the homeostasis of naïve T cells is well studied (*Puleston and Simon, 2014*), nothing is known about the requirement of autophagy in antigen-experienced T cells. The T cell memory pool has previously been shown to be controlled by PI3K/Akt and AMPK signaling as well as mTOR (mammalian Target Of Rapamycin) inhibition (*Araki et al., 2009*; *Kim et al., 2012*; *Rolf et al., 2013*), all of which also control autophagy (*Jung et al., 2010*). Indeed, T cell memory responses can be improved with the mTOR inhibitor rapamycin (*Araki et al., 2009*). Interestingly, rapamycin also rejuvenates HSCs in old mice (*Chen et al., 2009*). Neither of these studies addressed autophagy as a contributing mechanism.

**eLife digest** In the face of an infection, the immune system mounts an aggressive response by producing many copies of killer immune cells called CD8+ T cells that recognize and destroy any cells infected with the invading pathogen. The number of killer cells produced depends on the extent of the infection. Once the infection has been brought under control, most of the CD8+ T cells die off. The small numbers that are retained—called memory cells—'remember' the pathogen, so that if it invades the body again, they can help the immune system to respond more quickly and effectively.

Memory cells are also critical to the effectiveness of vaccines, many of which introduce a dead or weakened pathogen into the body. This does not cause an infection, but does allow the immune system to create memory cells that are able to fend off the same pathogen in the future. However, vaccines only work in individuals that are able to produce and maintain memory cells, which many older people are less able to do.

An important system that maintains cells, called autophagy, destroys and removes the 'junk' and toxic by products that all cells accumulate over time as a result of normal cell functions. Without autophagy, cells become less able to produce energy and they may die. Puleston et al. show that autophagy begins to fail in old mice, which prevents the formation of a proper memory response. In addition, mice that lack an important gene needed for autophagy are unable to produce memory cells after being infected with viruses such as influenza.

Puleston et al. found that boosting autophagy in older mice using a chemical called spermidine—which is also found naturally in many tissues—helped to restore the mice's ability to create and maintain memory cells. Spermidine-treated mice developed a stronger immunity to influenza after vaccination compared with other mice of a similar age. Further research is required to better understand how spermidine works to see if it could be developed into a drug that safely boosts the immune system of humans.

Evidence for age-related declining levels of autophagy stems largely from lower organisms such as yeast, flies, and worms (*Rubinsztein et al., 2011*). A recent study showed that the polyamine spermidine induces autophagy and thereby prolongs life span in model organisms (*Eisenberg et al., 2009*). Our own work demonstrated decreased levels of autophagy in CD8+ T cells of old individuals (*Phadwal et al., 2012*).

In this study, investigating a mouse model lacking the essential autophagy gene *Atg7* specifically in T cells, we find peripheral T cell lymphopenia, leading to proliferation and an activated phenotype within the CD8+ T cell compartment. While *Atg7*−/− T cells respond normally during the early stages of live viral challenge, a severely compromised memory CD8+ T cell compartment was found in response to influenza and murine cytomegalovirus (MCMV). Using bone marrow (BM) chimeras, we excluded that this is due the effects of lymphopenia; poor CD4+ T cell help; exhaustion, or altered cytokine receptor expression. Moreover, autophagy was found to be highest in antigen-specific CD8+ T cells when compared to naïve cells. Antigen-specific *Atg7*−/− CD8+ T cells also underwent more cell death at the time of memory formation, display compromised mitochondrial health, and increased expression of the glucose receptor GLUT1, a marker for glycolysis. Furthermore, recall CD8+ T cell responses to repeat immunizations and vaccination protocols were greatly diminished. This being reminiscent of the human ageing immune system (*Haq and McElhaney, 2014*), we confirmed reduced autophagy at the transcriptional and functional level in murine T cells from old mice. Importantly, we were able to restore the CD8+ T cell memory response in old mice with the autophagy-inducing compound spermidine, but not in autophagy-deficient mice. Finally, we found that spermidine induces autophagy independently of mTOR in T cells. Enhancing autophagy in an mTOR-independent manner may provide a safe way to improve vaccine responses in the elderly.

## Results

### Autophagy controls T cell numbers in naïve T-*Atg7*−/− mice

*Atg7flox/flox* mice were bred with *CD4-Cre* mice to generate mice with defective autophagy in both CD4+ and CD8+ T lymphocytes (T-*Atg7*−/−). Successful excision and thereby absence of *Atg7* mRNA

and Atg7 protein was confirmed in purified T cells (*Figure 1—figure supplement 1A and B*, respectively). Using the imaging flow cytometer (ImageStream) to count LC3 puncta in CD4$^+$ and CD8$^+$ T cells (*Phadwal et al., 2012*), we demonstrated that functional autophagy was significantly diminished in *Atg7$^{-/-}$* CD8$^+$ T cells (*Figure 1—figure supplement 1C* with examples of ImageStream images in right panel). In addition, using a classical technique to detect lipidated LC3, we confirmed that basal autophagy was diminished in the presence and absence of the autophagy flux inhibitor Bafilomycin A (*Figure 1—figure supplement 1D*).

Previous reports have noted a number of changes to the naïve CD8$^+$ T cell compartment in the absence of autophagy, with T cell lymphopenia, a consistent observation (*Pua et al., 2007*; *Puleston and Simon, 2014*). We set out to investigate if an altered naïve CD8$^+$ T cell compartment exists in T-*Atg7$^{-/-}$* mice. We confirmed observations from previous reports using similar autophagy-deficient mouse models (*Pua et al., 2007*, *2009*) that thymic development of CD4$^+$ and CD8$^+$ T cells was normal in 6-week old T-*Atg7$^{-/-}$* mice (*Figure 1A*). However, mice were lymphopenic for both CD4$^+$ and CD8$^+$ T cells in the lymph nodes and blood (*Figure 1B,C*). Moreover, *Atg7$^{-/-}$* CD8$^+$ T cells exhibited an activated phenotype with increased CD44 expression (*Figure 1D*) and decreased CD62L expression (*Figure 1E*), resembling a 'virtual memory' compartment (*Akue et al., 2012*). We observed similar frequencies of central effector memory CD62L$^+$CD44$^{hi}$, however, T-*Atg7$^{-/-}$* mice accumulated CD8$^+$ T cells with an effector memory phenotype (CD62L$^-$CD44$^{hi}$) (*Figure 1—figure supplement 1E*). Similarly, data from a mouse model in which another essential autophagy gene, *Atg5*, was deleted under the hematopoietic stem cell-specific promoter Vav, showed T cell lymphopenia and the expanded virtual memory T cell compartment (CD8$^+$CD44$^+$) suggesting this phenotype is not *Atg7* specific (*Figure 1—figure supplement 2A and B*). Next, we established that proliferation was increased in the activated CD44$^{hi}$ CD8$^+$ T cell compartment by Ki-67 staining (*Figure 1F*). The observed activated phenotype and increased cell turnover in *Atg7$^{-/-}$* CD8$^+$ T cells are likely driven by homeostatic proliferation in an attempt to fill the depleted T cell niche. Indeed, the expression of the homeostatic proliferation marker CD24 (*Li et al., 2006*) was found to be significantly increased on *Atg7$^{-/-}$* CD8$^+$ T cells (*Figure 1G*). To investigate whether lymphopenia drives this activated phenotype in the CD8$^+$ T cell compartment, we generated 1:1 mixed bone marrow (BM) chimeras from CD45.2$^+$ T-*Atg7$^{-/-}$* BM mixed with CD45.1$^+$ wild-type BM. Both BMs contributed equally to form the new hematological system (*Figure 1—figure supplement 2C*). *Atg7$^{-/-}$* CD8$^+$ T cells were still diminished even in the presence of a replete T cell niche, suggesting the survival defect previously described for autophagy-deficient T cells (*Pua et al., 2007*; *Mortensen et al., 2010*) is cell-intrinsic (*Figure 1—figure supplement 2D*). However, the activated *Atg7$^{-/-}$* CD8$^+$ T cell phenotype was no longer detected in BM chimeras as measured by the frequency of donor CD45.2$^+$ CD8$^+$ T cells found to be CD62L$^+$ (*Figure 1H*) and CD44$^{hi}$ (*Figure 1I*). These data indicate that the observed homeostatic proliferation and the change in surface phenotype of *Atg7$^{-/-}$* CD8$^+$ T cells are driven by lymphopenia and are not cell-intrinsic.

## Autophagy controls the CD8$^+$ T cell memory pool

Previous studies have shown a role for autophagy in naive T cell organelle homeostasis and survival (*Pua et al., 2009*; *Jia and He, 2011*). However, the importance of autophagy in CD8$^+$ T cells responding to infection is unknown. To investigate this, we challenged T-*Atg7$^{-/-}$* mice with influenza (PR8 strain) and MCMV. In T-*Atg7$^{-/-}$* mice, we found normal expansion of the antigen-specific effector CD8$^+$ T cell (CD8$^+$ T$_{eff}$) compartment using influenza-specific tetramers on day 10 (peak response) in the lungs (*Figure 2A*). However, due to the pre-existing lymphopenia observed in naïve mice, the absolute counts were diminished (*Figure 2—figure supplement 1A*). The CD8$^+$ T$_{eff}$ response to MCMV in the blood on day 7 was also normal (*Figure 2B*) (*Hutchinson et al., 2011*). However, the ability of T-*Atg7$^{-/-}$* mice to form memory CD8$^+$ T cells (CD8$^+$ T$_{mem}$) to both influenza and MCMV is severely compromised (*Figure 2C,D*). Performing serial bleeding on influenza infected mice over time, we demonstrated a catastrophic collapse of the antigen-specific CD8$^+$ T cell pool at day 21, resulting in a failure to retain CD8$^+$ T$_{mem}$ in T-*Atg7$^{-/-}$* mice (*Figure 2E*). Interestingly, the response to the 'inflationary' epitopes m38 and IE3 from MCMV is also compromised (*Figure 2F*). In response to these epitopes, wild-type CD8$^+$ T cells continue to expand throughout MCMV chronic infection. However, *Atg7$^{-/-}$* CD8$^+$ T cells fail to inflate and instead undergo a dramatic contraction leading to a failure to form a MCMV-specific CD8$^+$ T cell memory pool. This profound CD8$^+$ T cell contraction in T-*Atg7$^{-/-}$* mice is also observed in response to a conventional, non-inflating MCMV epitope (m45, *Figure 2—figure supplement 1B*). While viral titers against influenza in the lungs of wild-type and T-*Atg7$^{-/-}$* mice were comparable at day

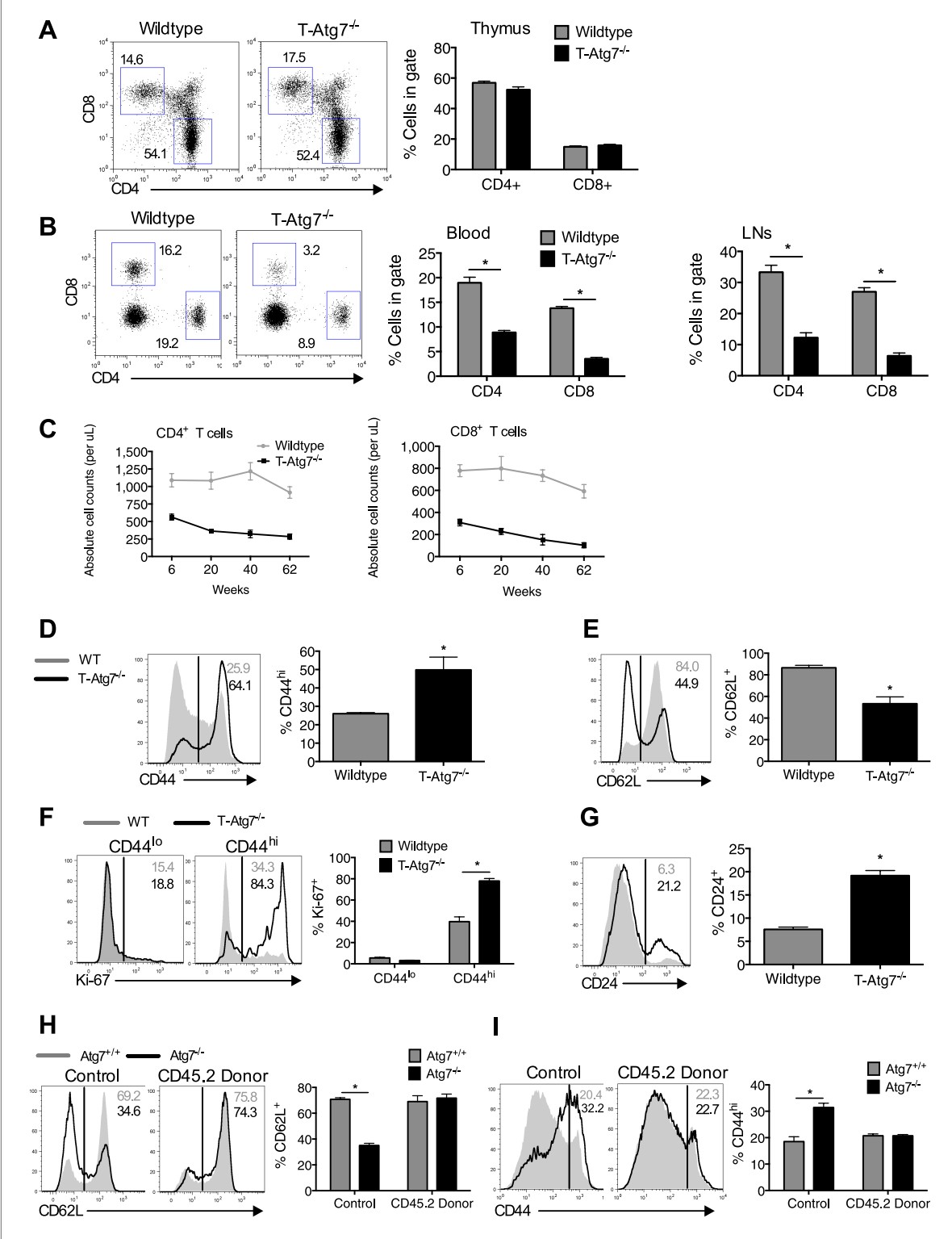

**Figure 1**. Lymphopenia induces homeostatic proliferation and an activated CD8[+] T cell phenotype in T-Atg7[−/−] mice. (**A**) Frequency of mature single positive CD4[+] and CD8[+] T cells in thymi of 6 week old mice (n = 4), representative FACS plot of three independent experiments. (**B**) Flow cytometric analysis of CD4[+] and CD8[+] T cell frequencies in blood and lymph nodes of T-Atg7[−/−] and wild-type mice (dot plots depict staining in blood). Quantitative analyses are representative of six independent experiments, *p < 0.05 (n = 4). (**C**) Absolute counts of CD4[+] and CD8[+] T cells in blood of T-Atg7[−/−] and WT mice over time (n = 4). (**D**) Percentage of CD44[hi] cells in the splenic CD8[+] T cell compartment of T-Atg7[−/−] and WT mice. Bar graphs depict the

*Figure 1. Continued on next page*

*Figure 1. Continued*

frequency of gated cells (representative of seven independent experiments), *p < 0.05 (n = 4). (**E**) Percentage of splenic CD8+ T cells positive for CD62L in T-*Atg7*–/– and WT mice. Data are representative of three independent experiments, *p < 0.05, (n = 4). (**F**) Percentage of CD44$^{lo}$ and CD44$^{hi}$ CD8+ T cells expressing Ki-67 in the spleen of T-*Atg7*–/– and WT mice. Bar graphs are representative of three independent experiments, *p < 0.05 (n = 4). (**G**) Frequency of splenic CD8+ T cells expressing CD24 in T-*Atg7*–/– and WT mice. Bar graph is representative of two independent experiments, *p < 0.05 (n = 4). (**H**) Percentage of donor-derived CD62L+CD8+ T cells. Lethally irradiated CD45.1 hosts reconstituted with a 1:1 mix of T-*Atg7*–/– or WT BM (both CD45.2) with CD45.1 wild-type BM. Controls were WT and T-*Atg7*–/– mice; *p < 0.05 (n = 4). (**I**) Frequency of donor-derived CD45.2+ CD8+ T cells in the spleen that is CD44$^{hi}$. Controls were normal WT and T-*Atg7*–/– mice, *p < 0.05 (n = 4). All values are mean ± s.e.m, and all statistical analyses are Mann–Whitney U-tests.

The following figure supplements are available for figure 1:

**Figure supplement 1**. *Atg7* is efficiently excised in CD8+ T cells from T-*Atg7*–/– mice resulting in loss of functional autophagy.

**Figure supplement 2**. The T cell phenotype of Vav-Atg5–/– mice is similar to that of T-*Atg7*–/– mice and the survival defect of Atg7–/– CD8+ T cells is cell intrinsic.

3 of infection, they were significantly higher on day 6 in T-*Atg7*–/– mice (***Figure 2G***). This is most likely due to the significantly lower absolute number of antigen-specific effector CD8+ T cells. As T-*Atg7*–/– mice survived influenza challenge, we expect that the virus is eventually cleared in all mice. In keeping with these findings, autophagy levels (CytoID) are significantly increased in the antigen-specific CD8+ T cell compartment compared to naïve T cells (CD44$^{lo}$) in response to influenza in both spleen (***Figure 2H***) and lungs (***Figure 2I***), indicating autophagy is induced upon antigen stimulation in vivo.

## The requirement for autophagy in memory CD8+ T cell differentiation is cell intrinsic

To exclude that a failure to maintain a CD8+ T$_{mem}$ pool is caused by lymphopenia in T-*Atg7*–/– mice, we performed viral challenge experiments in 1:1 mixed BM chimeras, facilitating the observation of antigen-specific *Atg7*–/– CD8+ T cell responses in a replete T cell environment. Even in the context of a BM chimera, *Atg7*–/– CD8+ T cells fail to generate a CD8+ T$_{mem}$ pool following influenza infection, suggesting this defect is cell intrinsic (***Figure 3A***). By using BM chimeras to observe antigen-specific *Atg7*–/– CD8+ T cell responses, we excluded a number of explanations as to why T-*Atg7*–/– mice fail to form CD8+ T$_{mem}$: (1) lymphopenia, (2) defective CD4+ T cell help, and (3) the excision of *Atg7* in CD4-expressing antigen-presenting cells affecting CD8+ T cell priming. Furthermore, a defect in the frequency of memory precursor cells might also impact upon CD8+ T$_{mem}$ formation in T-*Atg7*–/– mice. Mixed BM chimeras demonstrated that the frequency of short-lived effector (SLEC, CD127+KLRG1–) and memory precursor (MPEC, CD127–KLRG1+) CD8+ T cells was in fact normal in T-*Atg7*–/– mice (***Figure 3B***). We then tested whether the lack of memory response is due to exhaustion of autophagy-deficient antigen-specific CD8+ T cells. While in T-*Atg7*–/– mice, we found a sharp increase of antigen-specific CD8+ T cells co-expressing the exhaustion markers PD-1+ and TIM-3+; this was not found in the mixed BM chimeras. This indicates that without lymphopenia, exhaustion is not a hallmark of *Atg7*–/– CD8+ T cell responses and cannot explain the diminished CD8+ T$_{mem}$ compartment (***Figure 3C***). We then sought to determine if altered expression of cytokine receptors crucial for CD8+ T$_{mem}$ maintenance could explain the defective CD8+ T$_{mem}$ compartment. However, both Il-7Rα (CD127) and Il-15Rα were normally expressed on *Atg7*–/– MCMV-specific CD8+ T cells (***Figure 3D,E***).

Finally, we tested whether autophagy compromises effector/memory differentiation and measured two transcription factors typically required during this differentiation. We found that the expression of IRF4, responsible for sustaining the expansion and differentiation of CD8+ T$_{eff}$ (***Yao et al., 2013***), and EOMES, a factor that promotes CD8+ T$_{mem}$ formation (***Intlekofer et al., 2005***), behaved normally over time in *Atg7*–/– CD8+ T cells (***Figure 3—figure supplement 1A and B***).

## Autophagy is required to maintain healthy mitochondria and cell viability for antigen-specific CD8+ T cells

To understand the mechanism that leads to the failure to maintain a memory compartment in the absence of autophagy, we next checked if *Atg7*–/– CD8+ T cells were undergoing more cell death than wild-type cells. While this is well described for naïve autophagy deficient T cells (***Puleston and Simon, 2014***),

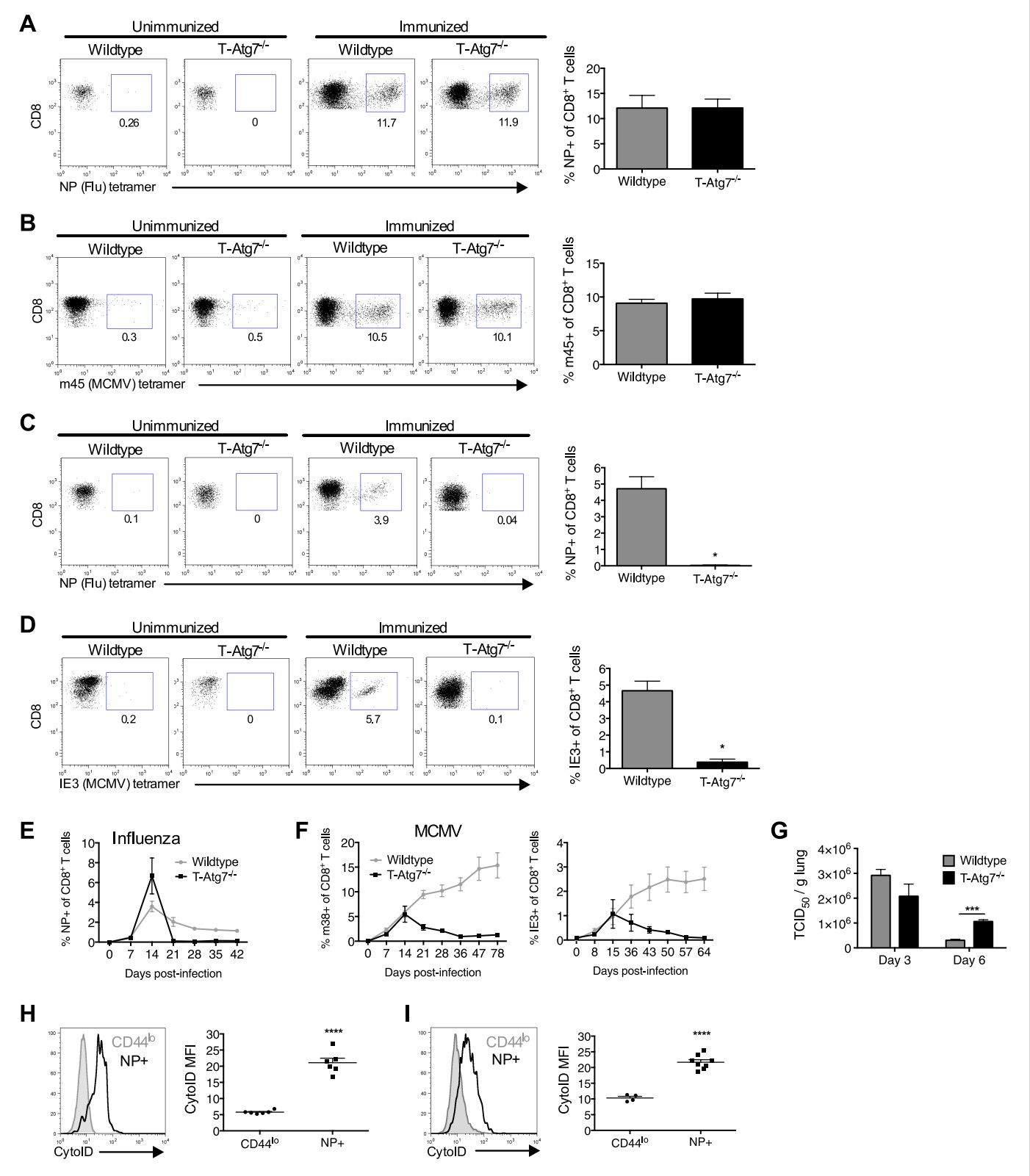

**Figure 2**. Normal effector CD8+ T cell responses to viral infection but defective memory CD8+ T cell formation in T-*Atg7*−/− mice. (**A**) Effector CD8+ T cell response to influenza in WT and T-*Atg7*−/− mice. Mice were immunized intra-nasally with 0.00032 HAU PR8 influenza. On day 10, antigen-specific CD8+ T cells to nucleoprotein (NP) was assessed with tetramer in lungs. Dot plots show examples of tetramer staining gated on CD8+ T cells. Bar graph

*Figure 2. Continued on next page*

*Figure 2. Continued*

indicates percentage of CD8+ T cells specific for NP (n = 5–6) and is representative of three independent experiments. (**B**) Effector CD8+ T cell's response to MCMV in WT and T-*Atg7*−/− mice. CD8+ T cells from blood were stained with m45 tetramer on day 7 post-infection. Dot plots show m45-tetramer+ cells gated on CD8+ T cells. Bar graph indicates % CD8+ T cells specific for m45 (n = 4–5) and is representative of three independent experiments. (**C**) CD8+ T$_{mem}$ response to influenza. WT and T-*Atg7*−/− mice immunized as in (**A**) and the antigen-specific CD8+ T cell response was assessed in lungs on day 50 by tetramer. *p < 0.05, by Mann–Whitney U-test (n = 4). Dot plots are gated on CD8+ T cells. Bar graph is representative of three independent experiments. (**D**) CD8+ T$_{mem}$'s response to MCMV. Lung CD8+ T cells on day 65 post-infection were stained with IE3-tetramer. Dot plots are gated on CD8+ T cells. Quantitation depicts frequency of IE3 specific CD8+ T cells. *p < 0.05, by Mann–Whitney U-test (n = 4). Data are representative of two independent experiments. (**E**) CD8+ T cell kinetics to influenza infection. WT and T-*Atg7*−/− mice were immunized as in (**A**) and CD8+ T cell response tracked over time in blood by tetramer. Y-axis shows frequency of NP-specific CD8+ T cells. (**F**) CD8+ T cell kinetics to MCMV infection. CD8+ T cell response to epitopes m38 (left panel) and IE3 (right panel) were tracked over time in blood by tetramer in WT and T-*Atg7*−/− mice. Y-axis indicates the percentage of CD8+ T cells that are m38-specific. (**G**) Influenza virus titres. WT and T-*Atg7*−/− mice were culled at days 3 and 6 post-immunization with PR8, and lungs were collected and snap frozen in liquid nitrogen. Virus titres were determined using MDCK-SIAT1 cells. ***p = 0.0002 as determined by Student t-test (n = 4) (**H**) WT mice were immunized with PR8 influenza as in (**A**). CD8+ T cells from spleen were stained with CytoID at day 9 post-infection and assessed by flow cytometry. Histograms show examples of CD44$^{lo}$ CD8+ T cells from unimmunized mice (filled grey line) and in NP-specific CD8+ T cells from immunized mice (open black line). Quantification is by mean fluorescence intensity (MFI) of CytoID on gated indicated cell population and representative of two independent experiments. ****p < 0.0001 by Student *t* test (n = 6). (**I**) Autophagy levels by CytoID staining on CD8+ T cells from the lungs on day 9 post-infection as in (**G**). Histograms provide examples of CytoID staining, gated on CD8+ T cells in lungs of immunized mice (solid grey lines), and gated on NP-tetramer-specific CD8+ T cells from immunized mice (open black lines). Quantification is by mean fluorescence intensity (MFI) of CytoID on gated indicated cell population and representative of two independent experiments. ****p < 0.0001 by Student *t* test (n = 4–8). All values are mean ± s.e.m.

The following figure supplement is available for figure 2:

**Figure supplement 1**. T-*Atg7*−/− mice fail to form memory CD8+ T cells to a conventional MCMV epitope.

it had not been analyzed in antigen-specific T cells. As expected, in wild-type mice challenged with MCMV, we found highest levels of cell death among antigen-specific CD8+ T cells just after the peak of the effector phase on day 9, whereas in T-*Atg7*−/− mice, cell death was found to increase over time (***Figure 4A***). This could not be explained by a change in levels of the anti-apoptotic protein Bcl-2 (***Figure 4—figure supplement 1A***). The control of mitochondrial quality and reactive oxygen species (ROS) via mitophagy, the degradation of mitochondria, has been found to prevent cell death in T cells (***Pua et al., 2009***). In keeping with this, mitochondrial content (***Figure 4B***) and mitochondrial ROS (***Figure 4C***) were significantly increased in *Atg7*−/− antigen-specific CD8+ T cells. While this likely explains the increased cell death, as shown for other hematopoietic cells (***Mortensen et al., 2010***) and T cells (***Pua et al., 2009***), mitophagy is also thought to be essential for maintenance of healthy mito-chondrial energy generation. This notion is consistent with our data in other *autophagy-deficient* hema-topoietic cell types (macrophages and primary leukemic lines, submitted) demonstrating increased glycolytic enzymes by proteomic analysis, increased lactate production and decreased oxygen con-sumption using metabolic seahorse measurements. Indeed, studies by Pearce et al showed that for-mation of the CD8+ T$_{mem}$ pool is accompanied by a switch to mitochondrial respiration (***Pearce et al., 2009***; ***Sukumar et al., 2013***). However, in the physiological viral infection models used here, aiming to mimic human infection, the scarcity of antigen-specific CD8+ T cells prevented us from performing these metabolic measurements. In addition, due to the loss of CD8+ T$_{mem}$ in the absence of *Atg7* by day 30, any metabolic analysis was restricted to early time points when antigen-specific CD8+ T cells are still present in T-*Atg7*−/− mice. We resorted to the measurement of a surrogate marker of glycolysis, the glucose transporter GLUT-1 on antigen-specific CD8+ T cells by two different techniques, (a) the fluorescently labeled GLUT-1 binding domain of HTLV (human T cell leukemia virus), which specifi-cally detects GLUT-1 (***Manel et al., 2003***; ***Kinet et al., 2007***), and (b) a GLUT-1 antibody staining. As expected, GLUT-1 is upregulated on CD8+ T$_{eff}$ (day 9) and then downregulated on CD8+ T$_{mem}$ (day 22) in wild-type mice as CD8+ T cells switch to mitochondrial respiration (***Figure 4D***). However, antigen-specific *Atg7*−/− CD8+ T$_{eff}$ expresses more GLUT-1 and downregulation does not occur to the same extent at the later time points compared to WT cells (day 22) (***Figure 4D***). This suggests that autophagy supports the metabolic switch during the differentiation from T$_{eff}$ to T$_{mem}$. This was confirmed using the GLUT-1 antibody (***Figure 4—figure supplement 1B***). However, treatment with the anti-diabetic drug metformin that induces mitochondrial β-oxidation metabolism in T cells (***Pearce et al., 2009***) did not rescue the memory T cell compartment in this model (data not shown).

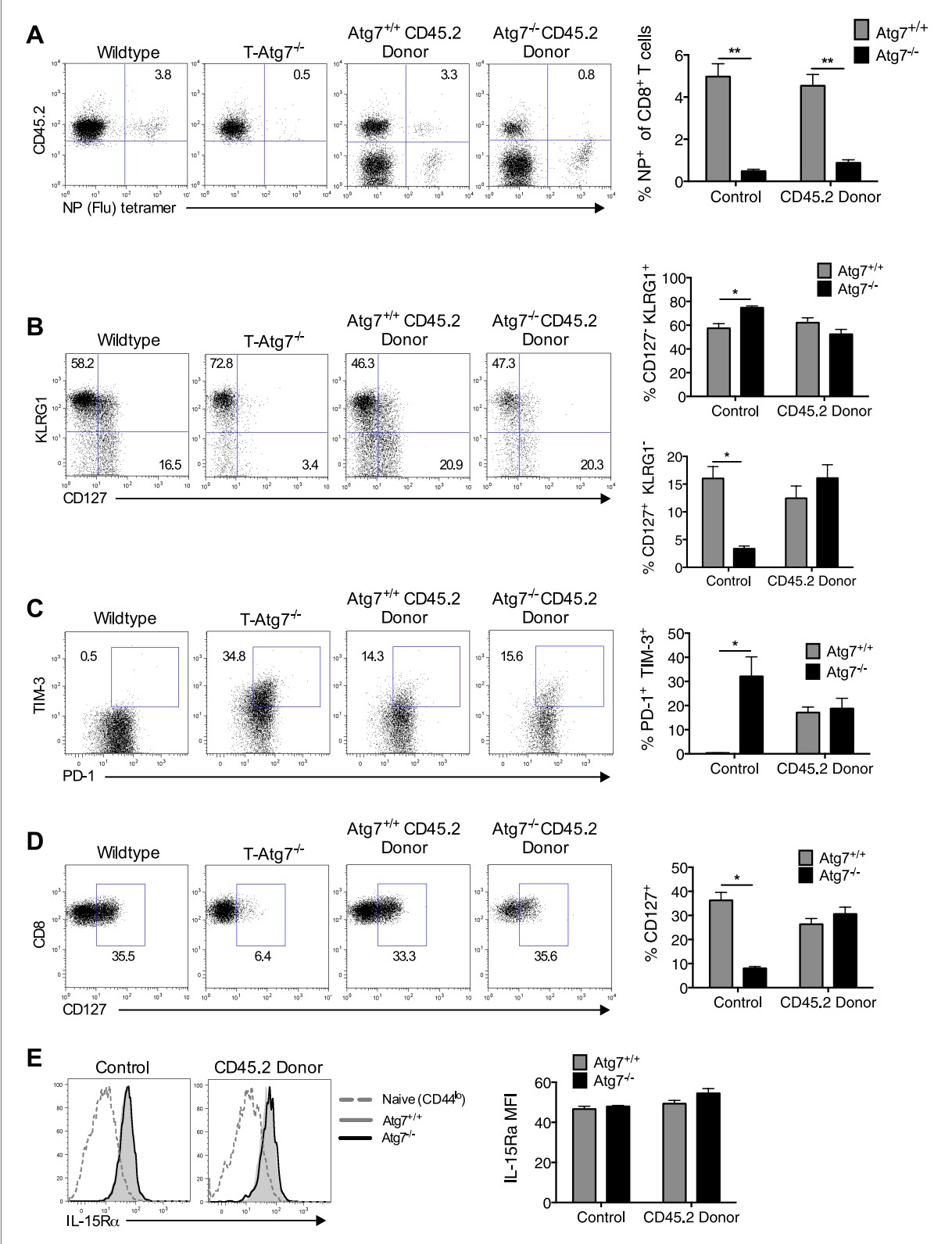

**Figure 3**. Loss of memory CD8+ T cell formation in the absence of *Atg7* is not due to lymphopenia, poor CD4+ T cell help, exhaustion, or defective cytokine receptor expression. (**A**) CD8+ Tmem response to influenza in mixed BM chimeras, generated as in *Figure 1H*. Mice were immunized with PR8 influenza, and the CD8+ Tmem response of the CD45.2 donor to NP was assessed in lungs by tetramer on day 40. Quantitation shows frequency of (donor)

*Figure 3. Continued on next page*

Figure 3. Continued

CD45.2$^+$ CD8$^+$ T cells that are NP-specific (n = 5–6). **p < 0.01, by Mann–Whitney U-test (n = 4–7). (**B**) SLEC and MPEC populations in the Atg7$^{+/+}$ and Atg7$^{-/-}$ antigen-specific CD8$^+$ T cell pool. Mixed BM chimera generated as in **Figure 1H**, were immunized with MCMV 8 weeks after transplantation. Dot plots show example of KLRG1 and CD127 expression on gated CD45.2$^+$ m45-tetramer$^+$ CD8$^+$ T cells on day 10 post-infection. Upper bar graph depicts the % of CD45.2$^+$ m45-tetramer$^+$ CD8$^+$ T cells that are CD127$^-$ KLRG1$^+$ (SLECs). Lower bar graph shows the % of CD127$^+$ KLRG1$^-$ (MPECs) in the same population. *p < 0.05, by Mann–Whitney U-test (n = 4–7). (**C**) Markers of exhaustion on Atg7$^{-/-}$ MCMV-specific CD8$^+$ T cells on MCMV challenged BM chimera. Dot plots depict example of PD-1 and TIM-3 staining on gated CD45.2$^+$ m45-tetramer$^+$ CD8$^+$ T cells. Bar graph quantifies the percentage of (donor) CD45.2$^+$ m45-tetramer$^+$ CD8$^+$ T cells that are PD-1$^+$ TIM-3$^+$ at day 10 post-infection. *p < 0.05, by Mann–Whitney U-test (n = 4–7). (**D**) CD127 expression on Atg7$^{-/-}$ MCMV-specific CD8$^+$ T cells in MCMV challenged BM chimeras. Examples of CD127 staining on gated CD45.2$^+$ m45-tetramer$^+$ CD8$^+$ T cells from spleen on day 10 post-infection are shown. *p < 0.05, by Mann–Whitney U-test (n = 4–7). (**E**) IL-15Rα expression on splenic Atg7$^{-/-}$ MCMV-specific CD8$^+$ T cells in MCMV challenged BM chimeras. Histograms depict IL-15Rα expression in CD44$^{lo}$ CD8$^+$ T cells from unimmunized mice (grey dotted line), Atg7$^{+/+}$ CD45.2$^+$ m45-tetramer CD8$^+$ T cells (grey filled line), and Atg7$^{-/-}$ CD45.2$^+$ m45-tetramer CD8$^+$ T cells (black line). The left histogram shows expression in control normal WT and T-Atg7$^{-/-}$ mice, the right histogram indicates staining in donor CD45.2$^+$ cells from BM chimera mice. Quantified is IL-15Rα mean fluorescence intensity on gated CD45.2$^+$ m45-tetramer CD8$^+$ T cells (n = 4–7). All values are mean ± s.e.m.

The following figure supplement is available for figure 3:

**Figure supplement 1**. Failure of to form memory CD8$^+$ T cells in T-Atg7$^{-/-}$ mice is not the result of defective IRF4 or EOMES expression.

## Autophagy deficiency leads to diminished recall responses

We then tested whether T-Atg7$^{-/-}$ mice were able to mount a recall CD8$^+$ T cell response to a secondary infection. We primed T-Atg7$^{-/-}$ mice with PR8 (H1N1) followed by heterologous challenge with the X31 strain of influenza (H3N2). By challenging with a heterologous virus strain expressing different surface antigens, it is possible to significantly diminish the influence of antibodies in mediating immunity to secondary infection. Thus, heterotypic immunity relies heavily on cross-reactive CD8$^+$ T cell responses (**Zweerink et al., 1977**; **Townsend et al., 1986**), as opposed to homotypic immunity (for example PR8 primed, PR8 challenged) to which influenza-specific antibodies contribute (**Townsend et al., 1986**; **Epstein and Price, 2010**). While the wild-type mice mounted a fast and strong recall response upon X31 challenge in PR8-primed mice (PR8 + X31), this was drastically diminished in T-Atg7$^{-/-}$ mice in the lungs on day 5 (**Figure 5A**, absolute counts in **Figure 5—figure supplement 1A**). Even when three live virus immunizations were administered (PR8 primed, challenged with X31 then with PR8), T-Atg7$^{-/-}$ mice were unable to mount recall responses. Surprisingly, T-Atg7$^{-/-}$ mice survived the heterotypic viral challenges despite the vastly diminished numbers of secondary CD8$^+$ T$_{eff}$.

Using an established non-replicating pseudotyped influenza vaccination protocol (**Powell et al., 2012**), we next tested whether H1N1 vaccinated T-Atg7$^{-/-}$ mice could mount secondary CD8$^+$ T responses to heterotypic viral challenge. Mice were vaccinated twice, 22 days apart then challenged with X31 influenza. Large numbers of influenza NP-specific CD8$^+$ T cells were detected in the lungs of vaccinated wild-type mice 5 days post-challenge, but not in T-Atg7$^{-/-}$ mice (**Figure 5B**). When the CD8$^+$ T cell kinetics to the vaccine regimen were followed over time in the blood of T-Atg7$^{-/-}$ mice, influenza-specific CD8$^+$ T cells were detected at normal frequencies to primary immunization but the secondary 'booster' vaccine was unable to induce an increase in NP-specific CD8$^+$ T cell frequency like it could in wild-type mice (**Figure 5C**). Similarly, vaccinated T-Atg7$^{-/-}$ mice failed to generate significant CD8$^+$ T cell responses to viral challenge in the blood (**Figure 5D**). As before, all mice survived the live viral challenge (data not shown). In summary, these data indicate that autophagy is required for the CD8$^+$ T cell recall response to either repeated immunizations with live virus or vaccination followed by live viral challenge. However, these results were not repeated in a mixed bone marrow chimera setting, making it difficult to exclude that lymphopenia, CD4$^+$ T cell help or CD4-expressing antigen-presenting cells might be responsible for defective recall response in T-Atg7$^{-/-}$ mice.

## Rejuvenation of aged vaccine responses through autophagy

We reasoned that a natural setting where these findings might be applicable is the aged organism, where reduced autophagy levels are found in many organs and cell types (**Rubinsztein et al., 2011**). Indeed, we have previously shown that autophagy levels are significantly reduced in the CD8$^+$ T cells of aged healthy human donors (>65 years) as compared to young donors (<30 years) (**Phadwal et al., 2012**). Taking results presented here so far into account, we hypothesized that these low levels may

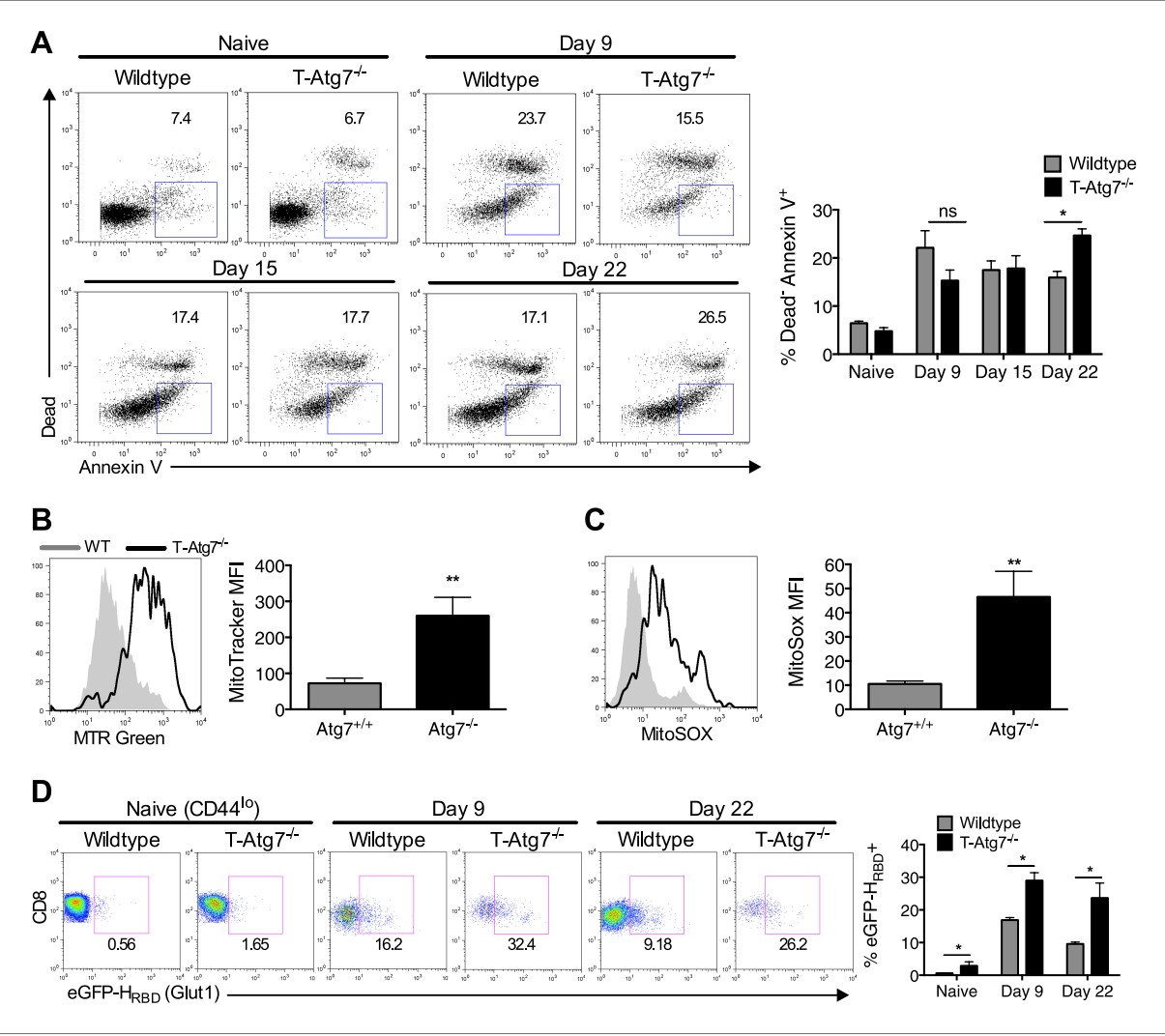

**Figure 4**. *Atg7−/−* memory CD8+ T cells show increased mitochondrial content, reactive oxygen species, apoptosis and fail to down-regulate GLUT-1.
(**A**) The spleens of unimmunized and MCMV-infected mice were stained with the apoptotic marker Annexin V and a dead cell dye that stains cells with disrupted membranes on the time points indicated. Apoptotic cells were defined as dead cell dye-negative Annexin V+. Dot plots are gated on either CD44lo CD8+ T cells (naïve) or m45-tetramer+ CD8+ T cells. *p < 0.05, by Mann–Whitney U-test (n = 4–5). (**B**) Mitochondrial volume by MitoTracker Green in Tetramer+ CD8+ T cells from WT and T-*Atg7−/−* mice. Spleens from MCMV-immunized mice were stained with MitoTracker Green on day 15 post-infection. Quantification depicts mean fluorescence intensity on m45-tetramer+ CD8+ T cells and is representative of three independent experiments. **p < 0.01, Student t test (n = 4–5). (**C**) Mitochondrial superoxide production in Tetramer+ CD8+ T cells by MitoSox. Spleens from MCMV-immunized WT and T-*Atg7−/−* mice were stained with MitoSox Red on day 15 post-infection and analyzed by flow cytometry. Bar graph depicts mean fluorescence intensity on m45-tetramer+ CD8+ T cells and is representative of three independent experiments. **p < 0.01, by Student t test (n = 4–5). (**D**) GLUT-1 expression. Tetramer+ CD8+ T cells from MCMV-immunized WT and T-*Atg7−/−* mice were stained with the GFP-tagged HTLV receptor binding domain (eGFP-HRBD), that binds GLUT-1, at the time points indicated. As a control, GLUT-1 was also measured on CD44lo CD8+ T cells from unimmunized mice (naïve). Bar graph shows the percentage of cells expressing GLUT-1. *p < 0.05, as determined by Mann–Whitney U-test (n = 4–5). All values are mean ± s.e.m.
The following figure supplement is available for figure 4:

**Figure supplement 1**. Normal Bcl-2 levels and altered GLUT-1 expression on antigen-specific *Atg7−/−* CD8+ T cells.

contribute to the poor CD8+ Tmem formation and inefficient influenza vaccination observed in the elderly (*Kapasi et al., 2002*; *Haynes et al., 2003*; *Kedzierska et al., 2012*).

We addressed whether diminished CD8+ Tmem formation can be improved in the elderly by modulating autophagy. First, we showed that mRNA levels of essential autophagy genes are decreased in sorted CD8+ T cells from naïve old mice (2 years) as compared to young mice (8 weeks). The

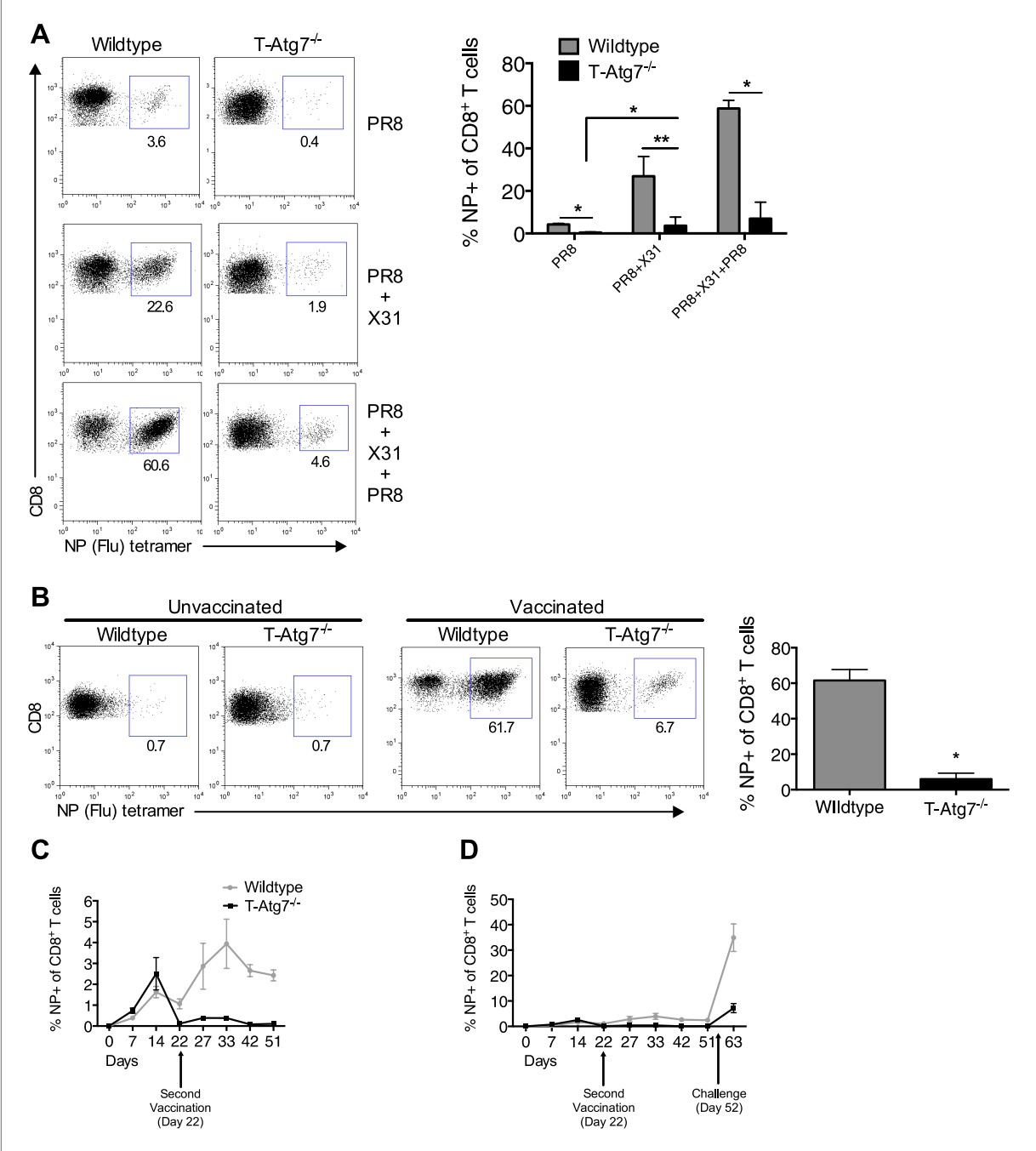

**Figure 5**. *Atg7⁻/⁻* memory CD8⁺ T cells mount a significantly reduced recall response to secondary immunization. (**A**) Recall responses to influenza. Here, WT and T-*Atg7⁻/⁻* mice were split into three groups. In one, mice were immunized with 0.00032 HAU PR8 influenza and the NP-specific CD8⁺ T cell response was measured in lungs on day 24 (PR8, n = 4). In another group, mice were immunized with 0.00032 HAU PR8 influenza followed by 0.32 HAU X31 influenza challenge on day 24. The recall response to this challenge was measured on day 5 post–challenge using NP-specific tetramers (PR8+X31, n = 6). In the third group, mice were immunized with 0.00032 HAU PR8 influenza, followed by 0.32 HAU X31 on day 24, and 30 days later immunized for a third time with 32 HAU PR8. The recall response to this third infection was measured in the lungs on day 5 post–challenge by tetramer (PR8 + X31 + PR8, n = 4). Quantification shows frequency of CD8⁺ T cells that are NP-specific and are representative of two independent experiments. *p < 0.05, **p < 0.01, by Mann–Whitney U-test. (**B**) The *Atg7⁻/⁻* CD8⁺ T cell response to influenza vaccination. WT and T-*Atg7⁻/⁻* mice were vaccinated twice, 22 days apart with 32 HAU of the live attenuated H1N1 vaccine, S-Flu. 30 days after the last vaccination, mice were challenged with 32 HAU X31 influenza. The CD8⁺ T cell response to NP was measured in lungs on day 23 post–challenge by tetramer. As a control, unvaccinated mice were challenged with 32 HAU X31 and culled on day 4 due to weight loss and morbidity. Quantification indicates the percentage of CD8⁺ T cells in the lung that are specific for NP on day

*Figure 5. Continued on next page*

*Figure 5. Continued*

23 post–challenge. Example dot plots are shown. Data are representative of two independent experiments. *p < 0.05 by Mann–Whitney U-test (n = 4–5). (C) *Atg7*$^{-/-}$ CD8$^+$ T cell kinetics to influenza vaccination. WT and T-*Atg7*$^{-/-}$ mice were vaccinated with S-Flu twice, 22 days apart, and the CD8$^+$ T cell response to NP was tracked over time in blood by tetramer (n = 4–6). (D) *Atg7*$^{-/-}$ CD8$^+$ T cell kinetics to influenza vaccination and challenge. WT and T-*Atg7*$^{-/-}$ mice were vaccinated as described in (C). 30 days after the last vaccination, mice were challenged with 32 HAU X31 influenza. Using NP-specific tetramers, the CD8$^+$ T cell response to the vaccine regime and the challenge was tracked in the blood over time (n = 4–6). All values are mean ± s.e.m.

The following figure supplement is available for figure 5:

**Figure supplement 1**. Atg7$^{-/-}$ CD8$^+$ T cells fail to mount robust re-call responses to secondary infection.

CD8$^+$ CD44$^{hi}$ memory compartment is particularly affected (*Figure 6A*). CD8$^+$ T cells from old mice also showed significantly decreased autophagic flux detected by counting LC3 spots in NP-specific CD8$^+$ T cells from young and old mice both in the presence and absence of an autophagy flux inhibitor (*Figure 6B,C*). This was confirmed by using two flow cytometry based autophagy detection, also in NP-specific CD8$^+$ T cells (*Figure 6—figure supplement 1A and B*).

Next, we chose the autophagy-inducing, naturally occurring compound spermidine to modulate autophagy in vivo, as it had been safely and effectively administered to mice previously (*Eisenberg et al., 2009*). We first confirmed that spermidine induces autophagy in T cells in vitro in a dose and time-dependent manner shown by increased levels of LC3-II (*Figure 6D*). Spermidine levels in the blood are known to decrease with age (*Pucciarelli et al., 2012*). To mimic this, we used the inhibitor of the natural spermidine synthesis pathway DFMO in the human Jurkat T cell line and found that autophagy levels detected by LC3 western blot significantly dropped in BafA treated samples and untreated samples (*Figure 6E*). Autophagy levels could be rescued by the addition of very low levels of spermidine in T cells (*Figure 6E*). As mTOR inhibition, such as achieved with rapamycin, is well known for its multiple side effects (*Lamming et al., 2013*), we confirmed that spermidine induces autophagy in T cells in an mTOR-independent manner. Rapamycin completely abolished the phosphorylation of S6K, a major target downstream of activated mTOR, whereas spermidine had no such effect (*Figure 6F*). Finally, we administered spermidine to old and young mice via the drinking water at concentrations known to induce autophagy (*Eisenberg et al., 2009*) during the influenza vaccination protocols as in *Figure 5B–D*. In response to first and second vaccination, and as observed in ageing humans, old mice do not mount a robust CD8$^+$ T cell response as compared to young mice. Spermidine dramatically enhanced influenza-specific CD8$^+$ T cell responses in old vaccinated wild-type mice but not in old vaccinated T-*Atg7*$^{-/-}$ mice as tracked over time in blood (*Figure 6G*). Similarly, the response to live influenza challenge in the blood was improved twofold to threefold (back to levels similar to young mice) when spermidine was administered to old mice, but not in the absence of autophagy (*Figure 6H*). In the lungs, the number of NP-specific CD8$^+$ T cells responding to influenza challenge in vaccinated old mice were improved fivefold to sixfold in the presence of spermidine, but had little effect on the response of aged T-*Atg7*$^{-/-}$ mice (*Figure 6I*).

## Discussion

Hematopoietic and immune health both decline significantly with age (*Beerman et al., 2010*). With the ageing population, this has become a substantial health and socio-economic problem for the developed world. Our work sheds light on the cellular mechanisms of immune ageing and how to improve responses to vaccination in the elderly.

The naïve T cell compartment in the absence of several different essential autophagy genes has been extensively analysed (*Puleston and Simon, 2014*). These models uniformly demonstrated increased damaged mitochondria leading to cell death and peripheral lymphopenia. In this study, we show how this lymphopenia drives the remaining T cells to take on an activated/memory phenotype as they proliferate in an attempt to re-fill the T cell niche, often termed 'virtual memory' (*Jia and He, 2011*). Thus, we provide a link between lymphopenia and the formation of the 'virtual memory' compartment and highlight the significant changes in T cell surface phenotype that can occur in a lymphopenic environment. Interestingly, the virtual memory compartment is also a hallmark of the ageing adaptive immune system. This observed phenotype in naïve mice would have precluded a coherent analysis of

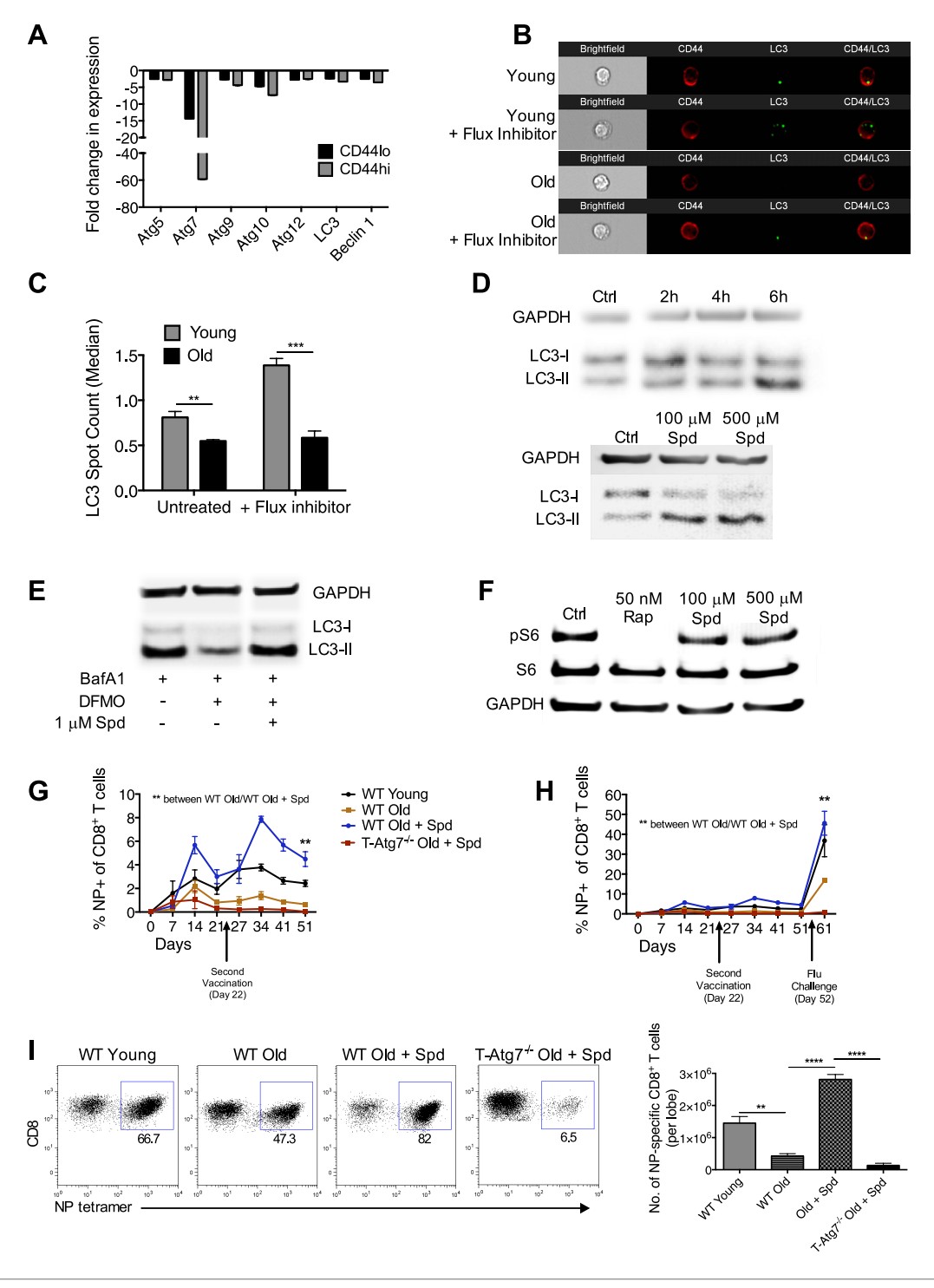

**Figure 6**. Boosting autophagy restores CD8+ T cell responses to vaccination in elderly mice. (**A**) Autophagy gene expression in CD8+ T cells from young and elderly mice. CD44lo and CD44hi CD8+ T cells were purified from 6 week old and 2 year old mice using fluorescent activated cell sorting. mRNA was extracted and the expression of essential autophagy genes was measured by q-PCR. Shown is the fold change in expression in CD8+ T cells from old mice relative to expression in young mice (normalized to *gapdh* and *hprt*). (**B**) LC3 Spot count in CD8+ T cells from young and old mice. Splenic CD8+ T cells from 8 week old and 2 year old mice were treated with an autophagy flux inhibitor for 2 hr, as a control cells were left untreated. LC3 spot count was determined on CD8+ CD44hi T cells

*Figure 6. Continued on next page*

*Figure 6. Continued*

using ImageStream. Representative images are shown (×60 magnification). **p = 0.0082, ***p = 0.0004 as determined by Student t-test (n = 4–5). (**C**) Quantification for images shown in (**B**). (**D**) Human T cell line Jurkat was incubated with 100 µM spermidine for 2, 4, or 6 hr or left untreated followed by whole protein extraction for LC3 Western Blot (upper panel). In the lower panel, Jurkat cells were incubated either with no spermidine (control), 100 µM, or 500 µM spermidine for 6 hr and then Western blotted for LC3. GAPDH was used as loading control for all Western blots. (**E**) Jurkat cells were treated with 1 mM DFMO or 1 mM DFMO with 1 µM spermidine for 48 hr or left untreated (control). In the final 6 hr of incubation, all cells were treated with 10 nM bafilomycin A1 and LC3-I to LC3-II conversion was assessed by Western Blot. (**F**) Jurkat cells were treated with 50 nM rapamycin, 100 µM or 500 µM spermidine for 6 hr followed by detection of phosphorylated S6 (Ser235/236) by Western Blot. As a control, cells were left untreated. (**G**) CD8$^+$ T cell kinetics to influenza vaccination in aged mice in the presence of spermidine. 8-week-old young WT and 23 month old WT and T-Atg7$^{–/–}$ mice were vaccinated 22 days apart with S-Flu. 21 days prior to the first vaccination, aged WT and T-Atg7$^{–/–}$ mice were administered spermidine in the drinking water at a concentration of 5 mM through to the experimental endpoint. As a control, 23 month old WT mice were administered water alone. The CD8$^+$ T cell response to NP was tracked over time in the blood by tetramer (n = 4–5). Y-axis depicts the frequency of CD8$^+$ T cells that are specific for NP. **p < 0.01 by Mann–Whitney U-test. (**H**) CD8$^+$ T cell kinetics to influenza vaccination and challenge in aged mice in the presence of spermidine. 8-week-old young WT and T-Atg7$^{–/–}$ and 23 month old WT and T-Atg7$^{–/–}$ mice were vaccinated as described in (**G**). 30 days after the last vaccination, mice were challenged with 32 HAU X31 and the CD8$^+$ T cell response to challenge was measured 9 days later in the lungs by tetramer. From 21 days prior to the first vaccination, through to the experimental end point, aged WT and T-Atg7$^{–/–}$ mice were administered spermidine as before. Y-axis indicated the percentage of CD8$^+$ T cells that are specific for NP (n = 4–5). (**I**) CD8$^+$ T cell response to influenza challenge in vaccinated aged mice in the presence of spermidine. 9 days post–challenge, lungs were harvested and the NP-specific CD8$^+$ T cell response to influenza challenge was measured by tetramer. Example dot plots are gated on CD8$^+$ T cells. Bar chart shows absolute counts of NP-specific CD8$^+$ T cells in the lung per lobe. **p < 0.01, ****p < 0.0001 by Student t test (n = 4–5). All values are mean ± s.e.m.

The following figure supplement is available for figure 6:

**Figure supplement 1**. Autophagy levels are significantly diminished in antigen-specific CD8$^+$ T cells from aged mice.

---

antigen-activated T cells; however, the use of mixed BM chimeras crucially excluded a contribution of the skewed naïve repertoire to the CD8$^+$ T$_{mem}$ phenotype.

In this study, we showed a novel and essential link for autophagy in the formation of CD8$^+$ T$_{mem}$ to infection. The absence of autophagy also dramatically altered the ability of CD8$^+$ T cells to respond to secondary infection. Interestingly, the severely depleted CD8$^+$ T cell recall response did not prevent T-Atg7$^{–/–}$ mice from surviving a lethal secondary heterotypic influenza challenge. Influenza virus is typically controlled by both antibodies and cellular immunity. However, in a heterotypic challenge, the surface hemagglutinin and neuraminidase (recognized by antibodies) differ between the two heterotypic strains, while the viral internal proteins (presented by MHC class I to CD8$^+$ T cells) remain the same, leaving the CD8$^+$ T cells to provide protection (*Rimmelzwaan et al., 2007*). The unique protective role of CD8$^+$ T cells in heterotypic immunity was described several decades ago in mice (*Rimmelzwaan et al., 2007*). The observation that the majority of CD8$^+$ T cell epitopes are cross-reactive between subtypes and are located in the most conserved regions of the internal proteins (nucleoprotein and matrix protein) confirms their role. In humans, the evidence, typical for human studies, is sparse and more circumstantial. A study from 1983 by McMichael et al demonstrated that in experimentally infected individuals, virus-specific cytotoxicity inversely correlated with the extent of virus shedding in the absence of virus-specific antibodies for the strain that was used for infection (*McMichael et al., 1983*). More recently, Lalvani et al found higher frequencies of pre-existing T cells to conserved epitopes in individuals who developed less severe illness in the 2009 H1N1 flu pandemic in the absence of cross-protective antibodies (*Sridhar et al., 2013*). We would postulate therefore that the survival of the T-Atg7$^{–/–}$ mice after the secondary heterotypic challenge relies on the very few effector CD8$^+$ T cells generated. These data also suggest that eliciting CD8$^+$ T cell memory responses in influenza vaccination is highly desirable, particularly to protect from pandemic flu.

This also raises interesting questions as to why such robust secondary T cell responses have evolved when survival can be achieved with a fraction of what is observed normally in nature. Although no CD8$^+$ T$_{mem}$ cells can be detected by conventional tetramer technology 20–30 days post-immunization

in T-*Atg7*⁻/⁻ mice, some memory cells must remain below the detection limit at very low frequency. In addition, these *Atg7*⁻/⁻ CD8⁺ T cells must maintain normal functional and proliferative capacity as they are detected at early time points after secondary immunization, characteristic of true memory T cell kinetics.

To explain how autophagy affects the maintenance of CD8⁺ $T_{mem}$ cells, we addressed two alternative hypotheses experimentally: (1) accumulation of cellular damage affects the long-lived CD8⁺ $T_{mem}$ in particular and (2) autophagy is required for the switch from glycolysis to mitochondrial respiration important for the survival of CD8⁺ $T_{mem}$ cells. Regarding the first one, we show evidence for the accumulation of mitochondria and ROS in the absence of autophagy. However, a limitation of our study is, we did not test for accumulation of other organelles and protein aggregates. As for the second hypothesis, initial studies involved TRAF6 (TNFR-associated factor 6) in the regulation of CD8⁺ $T_{mem}$ development by modulating fatty acid metabolism and thereby increasing mitochondrial respiration (*Pearce et al., 2009*; *van der Windt et al., 2012*). Interestingly, TRAF6, as an E3 ubiquitin ligase, has recently been shown to stabilize essential proteins in the autophagy pathway such as Ulk1 (*Nazio et al., 2013*) and Beclin-1 (*Shi and Kehrl, 2010*). We therefore hypothesize that TRAF6 controls metabolism via autophagy and propose three mechanisms contributing to autophagy's role in mitochondrial respiration: (1) through lipophagy, the degradation of lipids (*Weidberg et al., 2009*), autophagy provides the fatty acids that fuel the Krebs cycle with Acetyl-CoA though β-oxidation, (2) through degradation of glycolytic enzymes (*Xiong et al., 2011*), and (3) through mitophagy, autophagy provides mitochondrial quality control that is required for functional mitochondrial ATP generation. We provide evidence for the latter mechanism here. This suggests that autophagy is important in the control of metabolic pathways in T cells. Interestingly both the accumulation of damaged mitochondria/oxidative stress as well as a bias to preferentially use glycolysis are hallmarks of the ageing cell (*Hipkiss, 2006*).

In this study, we showed that autophagy levels are impaired in CD8⁺ T cells from aged mice. Using spermidine, we could dramatically improve the CD8⁺ T cell response to vaccination and infection in elderly mice in an autophagy-dependent manner. However, aged CD4⁺ T cells also undergo immune senescence. While inducing autophagy with spermidine in the CD8⁺ T cell compartment clearly contributes to the increased numbers of antigen-specific CD8⁺ T cells, the limitation of these experiments is that we cannot gauge the contribution of improved CD4⁺ T cell function to CD8⁺ T cell numbers. This will be included in future studies.

These results offer the first cell-intrinsic explanation as to why T cell memory is defective in old age. Spermidine is a pleiotropic compound and its major function has been described in yeast as an inhibitor of histone acetyl transferases, inducing epigenetic changes to autophagy-related gene expression (*Eisenberg et al., 2009*). We showed that spermidine operates independently of mTOR to induce autophagy. These findings offer the prospect of improving vaccine T cell responses in the elderly through spermidine in an mTOR-independent fashion. However, no other ways of inducing autophagy were tested here. We expect that fasting or other mTOR-independent drugs such as resveratrol (*Morselli et al., 2009*; *Morselli et al., 2010*) may also be beneficial for CD8⁺ $T_{mem}$ maintenance. However, spermidine is significantly more attractive than previously published $T_{mem}$-boosting compounds such as metformin (*Pearce et al., 2009*) and rapamycin (*Araki et al., 2009*), that come with unwelcome side effects and toxicity in humans. The elucidation of the precise role of spermidine will be the subject of future work and will enable its development as a safe, readily administrable immune-modulating drug.

## Material and methods

### Mice

*CD4-Cre* mice (*Lee et al., 2001*) were from Adeline Hajjar (University of Washington) and crossed with *Atg7*^flox/flox (*Komatsu et al., 2005*) to obtain T-*Atg7*⁻/⁻ mice (*CD4-Cre*⁺ *Atg7*⁺/⁺) on a C57BL/6 background. Unless otherwise stated, all mice were 6–12 weeks of age at the start of each experiment and were age and sex matched. *CD4-Cre*⁻ *Atg7*⁺/⁺ mice were used as wild-type controls and were littermates where possible. No phenotype was observed in *CD4-Cre*⁺ *Atg7*⁺/⁻ mice. Old mice and young control mice were purchased from Charles River, UK. C57BL/6 SJL CD45.1 mice for bone marrow chimeras were purchased from Biomedical Services, Oxford. All mice were housed in Biomedical Services, Oxford and animal experiments were approved by the local ethical review committee and performed under UK project license (PPL 39/2809).

## Flow cytometry

The following antibodies were used for flow cytometry (antibody clone in brackets): CD8 (Ly2) PE/PE-CY7; CD8 (53-6.7) FITC/eF450/PE-Cy7; CD4 (GK1.5) PE/FITC/APC; TCRβ (H57-597) FITC/PE/PE-Cy7/APC; CD3 (145-2C11) eF450/APC; CD62L (MEL-14) FITC/PE-Cy7; CD44 (IM7) FITC/PE-Cy7; KLRG1 (2F1) FITC/PE-Cy7; IRF4 (3E4) FITC; EOMES (Dan11mag) PE-Cy7; CD127 (A7R34) FITC/PE/eF450; Ki-67 (SolA15) eF450; CD24 (M1/69) APC; CD16/32 (93) purified Fc block; CD45.2 (104) eF450; all from eBioscience (San Diego, CA, USA). PD-1 (29F.1A12) PE-Cy7; PD-1 (RMP1-30) PE; CD45.2 (104) PE-Cy7; TIM-3 (B8.2C12) PE/APC; Bcl-2 (BCL/10C4) PE; all from Biolegend (San Diego, CA, USA). Glut1 (q) PerCP; IL-15Rα (BAF551) biotinylated; both from R&D Systems (Minneapolis, USA).

For surface staining, cells were suspended in PBS, 2% FCS, 5 mM EDTA and stained at 4°C. Anti-CD16/32 (Fc Block, eBioscience) was generally added to antibody mix to minimize non-specific staining. LIVE/DEAD Fixable Violet Dead Cell Stain Kit (Life Technologies) was used prior to surface staining to exclude dead cells. For cytoplasmic intracellular staining (Bcl-2), cells were stained for surface antibodies then fixed in IC fixation buffer before permeabilisation with perm buffer (eBioscience). Cells were then suspended in perm buffer at RT for antibody staining. For nuclear targets (IRF, EOMES, Ki-67), after surface staining cells were fixed and permeabilised with FoxP3 Fixation/Permeabilisation kit (eBioscience) before resuspension in perm buffer for intracellular staining at RT.

Absolute cell counts were performed on peripheral blood taken from the lateral tail vein of live animals, collected in heparin-coated tubes (Microvette 300, Sarstedt [Nümbrecht, Germany]) to avoid coagulation. Cell counts were calculated with BD TruCount tubes (BD Bioscience, NJ, USA) according to the manufacturer's instructions.

Tetramers were generated as previously described (*Altman et al., 1996*) and MHC-Class I monomers stored at −80°C. Biotinylated monomers were tetramerized with Streptavidin PE or APC at the right concentration to achieve 1:1 ratio with biotin binding sites and added in 1/10th volumes waiting 10 min between additions. Tetramerized complexes were stored at 4°C. Peptide sequences for MCMV tetramers were as follows: m45 $^{985}$HGIRNASFI$^{993}$, H-2D$^b$-restricted; m38 $^{316}$SSPPMFRV$^{323}$, H-2K$^b$-restricted; IE3 $^{416}$RALEYKNL$^{423}$, H-2K$^b$-restricted. The peptide sequence for the influenza tetramer was as follows: NP $^{366}$ASNENMETM$^{374}$, H-2D$^b$-restricted. Cells were always stained with tetramers prior to surface antibody staining in PBS, 2% FCS at 37°C for 15 min. Tracking tetramer-positive cells over time in the blood was performed by serially bleeding the lateral tail vein. After erythrocyte lysis with red cell lysis buffer (eBioscience), remaining white cells were stained with tetramer and surface antibodies as described above.

Autophagy detection by flow cytometry was measured using the CytoID Autophagy Detection Kit (Enzo Life Science, Exeter, UK). Splenocytes and cells isolated from the lung were first stained with CytoID according to the manufacturer's instructions, prior to tetramer and surface antibody staining. Autophagy was also measured using flow cytometry by quantifying LC3-II mean fluorescence intensity using the FlowCellect Autophagy LC3 Antibody-based Assay Kit (FCCH100171, Merk-Millipore, MA, USA) according to the manufacture's instructions and always following cell surface antibody staining. Use of this kit includes a step where cytosolic LC3-I is washed from the cell, leaving only membrane bound LC3-II prior to staining.

For apoptosis detection, splenocytes were stained with m45-tetramer and surface antibody as described above and then stained with LIVE/DEAD Fixable Violet Dead Cell Stain Kit. Cells were finally stained with Annexin V PE-Cy7 (eBioscience) in Annexin V binding buffer at room temperature. Apoptotic cells were determined as LIVE/DEAD cell dye negative (live cells), Annexin V positive.

GLUT-1 was measured through binding to its ligand, the receptor binding domain (RBD) of a recombinant glycoprotein from the human T lymphotrophic virus (HTLV) fused to eGFP (H$_{RBD}$-eGFP) (*Montel-Hagen et al., 2008*).

For mitochondrial mass analysis, cells were stained with MitoTracker Green (Life Technologies, Carlsbad, CA, USA) at 150 nM in PBS, 2% FCS for 30 min at 37°C after tetramer and surface antibody staining. To measure mitochondrial superoxide production, MitoSOX Red (Life Technologies) was used at final concentration of 5 μM from a 5 mM stock in PBS, 2% FCS for 30 min at 37°C following tetramer and cell surface staining. All flow cytometry experiments were performed on a Cyan flow cytometer (Beckman Coulter, Brea, CA, USA).

## Bone marrow (BM) chimera

BM cells were extracted from a single 8-week old wild-type, T-*Atg7*$^{−/−}$ (both CD45.2$^+$), and C57BL/6 SJL mouse expressing CD45.1. After erythrocyte lysis, $3 \times 10^6$ wild-type or T-*Atg7*$^{−/−}$ BM cells were

added to 3 × 10$^6$ CD45.1+ BM cells (1:1 CD45.2$^+$:CD45.1$^+$) in a total volume of 200 µl PBS. The 1:1 BM mix was injected i.v 2 hr after lethal irradiation (450 cGy twice, 4 hr apart) to C57BL/6 SJL CD45.1+ recipients (total of 6 × 10$^6$ BM cells per mouse in 200 µl). After 8 weeks of reconstitution, mixed BM chimera was immunized with MCMV or PR8 influenza, or left unimmunized. WT and T-Atg7$^{-/-}$ mice were always used as controls.

## Immunizations

Mice were immunized i.v. with 1 × 10$^6$ p.f.u MCMV in 100 µl (Smith strain ATCC: VR194). Unimmunized controls were injected with 100 µl PBS i.v. For influenza infections, mice were administered intra-nasally with either A/PR/8/34 (PR8, H1N1 Cambridge) influenza or X31 (H3N2) influenza at the stated dose in 50 µl viral dilution media (VDM; DMEM, 0.1% BSA, 10 mM HEPES, 100 U/ml penicillin, 100 µg/ml streptomycin, and 2 mM glutamine; all from Sigma, St Louis, MO, USA). Mice were anaesthetized with isofluorane and droplets of VDM-containing virus were applied to the nares until the total 50 µl was inhaled. Unimmunized mice were always used as controls (50 µl of VDM alone). For influenza vaccination, mice were immunized intra-nasally with 32 HAU pseudotyped H1N1 influenza (S-Flu) (*Powell et al., 2012*), using the influenza infection protocol described above, and challenged with 32 HAU X31. Unvaccinated and fully naïve mice were used as controls in all influenza vaccine experiments. In unvaccinated mice that received X31 challenge, a combination of weight loss and clinical score was used as a humane endpoint.

## q-PCR

Cells were purified with a MoFlo cell sorter (Beckman Coulter) by their surface markers. RNA was extracted using RNeasy Kit (Qiagen, Hilden, Germany) and quantified using a Nanodrop spectro-photometer (Thermo Scientific, Waltham, MA, USA). RNA was reverse transcribed (RT) using a High Capacity RNA to cDNA kit (Applied Biosystems (AB), Foster City, CA, USA). Resulting cDNA was stored at −20°C. Real-time quantitative PCR using comparative Ct method (ΔΔCt) was utilized to eval-uate gene expression using validated TaqMan probes (AB) on a 7500 Fast Real-time PCR machine (AB). Conditions: (1) 50°C, 2 min; (2) 95°C, 10 min; (3) 95°C 15 s; (4) 60°C 1 min; 40 cycles of 3–4. The assay IDs for the primers of the analyzed genes are as follows: Mm00504340_m1 (*Atg5*), Mm00512209_m1 (*Atg7*), Mm01264428_m1 (*Atg9*), Mm00470550_m1 (*Atg10*), Mm00503201_m1 (*Atg12*), Mm0051717_m1 (*beclin1*), Mm00458724_m1 (*Map1lc3a*), Mm01545399_m1 (*hprt*), Mm99999915_g1 (*gapdh*).

## ImageStream

ImageStream (Amnis imaging flow cytometer, MA, USA) has previously been used to determine autophagic flux (*Phadwal et al., 2012*). To determine LC3 spot count, we stained cells for LC3-II (following cell surface staining) using the FlowCellect Autophagy LC3 Antibody-based Assay Kit (FCCH100171, Merk-Millipore) according to the manufacture's instructions. Before LC3 detection, to assess autophagic flux, cells were incubated for 2 hr with the autophagy flux inhibitor provided by the kit specified and according to the manufacturer's instructions. LC3 spot count was quantified using Ideas software (Amnis), which contains a specialised objective spot counting feature. Images collected were all at ×60 magnification.

## Western blot

3 × 10$^6$ Jurkat cells (human CD4$^+$ T cell line) or 5 × 10$^6$ purified primary CD8$^+$ T cells were lysed on ice using 100 µl 1 × NP-40 Lysis Buffer. Protein concentration in supernatant was measured using BCA Protein Assay Kit (PI-23227, Thermo Scientific), and reducing Laemmli Sample Buffer was added to make protein samples for SDS-PAGE. Proteins of 30–50 µg per lane were separated on 16% SDS-PAGE and transferred to PVDF membrane (IPFL00010, Millipore, MA, USA). After blocking with 5% skim milk, the membrane was blotted using the following primary antibodies: LC3 (L8918, Sigma) (1:1000), GAPDH (MAB374, Millipore) (1:10,000), pS6 (2211, Cell Signaling Technology, Danvers, MA, USA) (1:5000). For S6 blotting, membranes were blotted with anti-pS6 first, then stripped for re-blot using primary antibody against S6 (2217, Cell Signaling Technologies) (1:1000). IRDye secondary antibodies were bought from LI-COR (Lincoln, NE, USA): (926-32,211, 926-68,020) (1:15,000). For the purification of primary CD8$^+$ T cells, cells were negatively selected from sple-nocytes using the CD8$^+$ T cell Isolation kit (130-104-075, Miltenyi, Bergisch Gladbach, Germany) according to the manufacturer's instructions.

## Influenza viral titers (TCID$_{50}$ and hemagglutination)

Determination of the 50% tissue culture infective dose (TCID$_{50}$) was performed on MDCK-SIAT1 cells in a 96-well flat-bottom plate performed by serial dilution of lung homogenates onto $3 \times 10^4$ MDCK-SIAT1 cells followed by incubation for 72 hr at 37°C. Virus was then detected by hemagglutination, where 50 µl 1% (vol/vol) human erythrocytes (adjusted so that a 1:2 dilution gave an optical density at 600 nm) was added to 50 µl of the serially diluted lung homogenates/MDCK-SIAT1 supernatant in a 96-well V-bottom plate. Hemagglutination was analyzed by the loss of teardrop formation after tilting the plate. TCID$_{50}$ was calculated as described by *Reed and Muench (1938)*.

## Statistical analyses

All data were presented as mean ± s.e.m. p values were determined using either a Mann–Whitney U-test or two-tailed Student's *t* test using GraphPad Prism software. Significant statistical differences are indicated in the figure legends or on the graphs themselves.

## In vivo spermidine treatment

Mice were administered spermidine in the drinking water 21 days prior to immunization through to the experimental endpoint. Spermidine was added at a concentration of 5 mM from a 1 M aqueous stock. Spermidine-containing drinking water was replenished every 2–3 days.

## Acknowledgements

The study was funded by the MRC Human Immunology Unit and the Wellcome Trust. The Chinese Scholarship Council is gratefully acknowledged for the stipend and fees of HZ; the Lady Tata fund for AW's stipend and fees; and TJP is funded by the Townsend-Jeantet Trust (registered charity no. 1011770) and the MRC Human Immunology Unit. Thanks also goes to Rosanna McEwen-Smith for assistance in setting up the influenza studies.

## Additional information

### Funding

| Funder | Grant reference number | Author |
| --- | --- | --- |
| Medical Research Council | Human Immunology Unit, Oxford | Daniel J Puleston, Hanlin Zhang, Timothy J Powell, Elina Lipina, Alexander S Watson, Vincenzo Cerundolo, Alain RM Townsend |
| Wellcome Trust | 088098/Z/08/Z | Daniel J Puleston, Paul Klenerman |
| Allan and Nesta Ferguson Charitable Trust | | Daniel J Puleston |
| Natural Sciences and Engineering Research Council of Canada | | Alexander S Watson |
| Lady Tata Memorial Trust | | Alexander S Watson |

The funders had no role in study design, data collection and interpretation, or the decision to submit the work for publication.

### Author contributions

DJP, Conception and design, Acquisition of data, Analysis and interpretation of data, Drafting or revising the article; HZ, EL, IP, ASW, Acquisition of data, Analysis and interpretation of data; TJP, Conception and design, Acquisition of data, Analysis and interpretation of data, Contributed unpublished essential data or reagents; SS, Acquisition of data, Contributed unpublished essential data or reagents; VC, Analysis and interpretation of data, Contributed unpublished essential data or reagents; ARMT, PK, Conception and design, Analysis and interpretation of data, Contributed unpublished essential data or reagents; AKS, Conception and design, Analysis and interpretation of data, Drafting or revising the article

## Ethics

Animal experimentation: Animal experiments were approved by the local ethical review committee and performed under the project licence (PPL 39/2809) issued by the UK home office.

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
