## [Decision Letter]

Thank you for sending your work entitled “Autophagy as a critical regulator of CD8+ T cell memory and a route to improved vaccination in the elderly” for consideration at *eLife.* Your article has been favorably evaluated by Tadatsugu Taniguchi (Senior editor), a member of the Board of Reviewing Editors, and 2 expert reviewers in your field.

The Reviewing editor and the reviewers discussed their comments before we reached this decision, and the Reviewing editor has assembled the following comments to help you prepare a revised submission.

The reviewers and Reviewing editor all feel that, overall, this is an interesting, well-performed study addressing the role of autophagy and ageing effects on autophagy in viral-specific CD8 memory T cells. We think that this study has important potential implications for understanding the role of autophagy in a crucial basic aspect of immunology, T cell memory, and, as pointed out by the authors, has potential clinical implications for improving vaccine responses in the elderly.

The reviewers have made several suggestions to further improve the manuscript. We are enclosing a list of the major points. Some of the points request further text explanations or modifications of the data analysis. Other points suggest a need for some additional data; however, as a general principle, given the overall enthusiasm for the work and the editorial policies of *eLife*, if these points cannot be quickly addressed by new experimental data, we ask that you revise the text (Discussion) to acknowledge the limitations of the present study.

One exception to this principle is point #8; although we recognize that your lab has previously published studies using the BDS feature to measure autophagic flux in lymphocytes, we do believe that, prior to publication, this assay should be confirmed by at least one additional autophagic flux assay to more convincingly demonstrate a defect in autophagic flux in T/Atg7-/- cells. Another exception is point #9; the LC3 assays in Figure 6 are performed in the same cells; the difference in the blots for control cells suggests a technical problem that should be resolved prior to publication.

1) The authors have not directly examined the effects of memory CD8 T cells on viral clearance and/or protection against pathological lung damage during influenza virus. If the authors have data in the adoptive transfer experiments in Figure 3 evaluate the functions of wild-type and Atg7-/- memory T on viral clearance and/or lung damage, such data should be included in the manuscript. If they have not collected such data in their experiments, they should acknowledge this limitation.

2) The authors use a model of heterotypic immunity against influenza to avoid a model in which antibody-mediated responses are involved in protection against viral challenge. However, it is notable that mouse survival is not affected by autophagy deficiency in T cells. The reviewers ask whether the authors have tried to use higher titers of the viruses or lethal strains such as those described before (Virology 2011, 412:36-45). If antibodies are the main effectors against influenza and CD8 T cells are not sufficient or required for the protection against influenza challenge, this should be clarified by the authors. In addition, the authors should more fully discuss the data on the role of human CD8 T cell responses in protection against influenza challenge after vaccines. The argument that the findings in mice may have relevance to diminished vaccine responses to flu vaccines in elderly relies on the relevance of CD8 T cell memory responses to vaccine efficacy. The final paragraph mentions briefly that CD8 T cell responses are an important correlate of vaccine protection; however, this should be discussed more extensively in light of the lack of in-depth experimental evidence in this study in the mouse for a role of memory CD8 T cells in protection against disease caused by viral challenge.

3) T/Atg7-/- mice show defects in T cell development and a reduction of CD4 and CD8 T cells in the spleen and lymph nodes (J. Immunol. 182:4046-4055 and Figure 2). In Figure 1, the authors should show the percentage and total numbers of antigen-specific CD8 T cells among all cells in the lung, not just among gated CD8 T cells. The authors conclude that there is “normal expansion of the antigen-specific effector CD8+ T cell compartment”. However, a general lymphopenia in CD4 and CD8 T cells in these mice should be clearly presented in Figure 1, although a comparable fold of CD8 expansion was observed.

4) The reviewers feel that the interpretation of data in Figure 5 is complicated by lymphopenia in T/Atg7-/- mice as well as autophagy deficiency in the CD4 T cell compartment. This point needs to be addressed in the authors' response.

5) In Figure 6, the authors should consider immunodeficiency in aged CD4 T cells (Wakikawa et al., Exp Gerontol 34, 231-242; Cheung et al., Exp Gerontol 18, 451-460). Spermidine would target all cells types in vivo, including CD4+ T helper cells. The experiments performed do not distinguish between these possibilities.

6) The knockout of essential genes from the autophagic cascade entails a major cellular dysfunction due to accumulating, dysfunctional mitochondria (and perhaps other organelles and protein aggregates). Hence, the phenotype of Atg7 knockout cells is not just the result of autophagy inhibition but rather that of accumulating damage. This problem should be discussed. It would be even better to include some of the data that are mentioned as “not shown” with regard to the phenotype of mice lacking Atg5 in T cells.

7) The authors only use one autophagy inducer, spermidine, in their experiments. Hence, it is still possible that autophagy would be only one of the spermidine effects (knowing that spermidine that might mediate additional metabolic effects on polyamine metabolism) that accounts for the improvement of CD8 memory T cells. The paper would be more complete, if the authors used other autophagy inducers (such as resveratrol, fasting etc). At a minimum, the current limitation should be discussed.

8) Autophagic flux is only measured by the 'BDS feature', namely the co-localization of a lysosomal marker and LC3, measured by means of the Amnis Imagestream system. It is important to confirm these measurements (and in particular the efficiency of the ATG7 knockout) by immunoblot detection of the LC3 lipidation in the absence and presence of autophagy inhibitors.

9) The controls in Figure 6 look very different with regard to the proportion of LC3-I and LC3-II. This could be due to several different factors, including differences in basal autophagy conditions (density, nutritional factors) or sample processing (i.e. freeze-thawing samples will result in degradation of LC3-I which is more labile). The discrepancy in experimental conditions that results in such a notable discrepancy in the proportion of LC3-I and LC3-II in controls of the same cell type in different figures should be resolved.

---

## [Author Response]

*1) The authors have not directly examined the effects of memory CD8 T cells on viral clearance and/or protection against pathological lung damage during influenza virus. If the authors have data in the adoptive transfer experiments in*
Figure 3
*evaluate the functions of wild-type and Atg7-/- memory T on viral clearance and/or lung damage, such data should be included in the manuscript. If they have not collected such data in their experiments, they should acknowledge this limitation*.

We have measured viral titers against influenza in the lungs of normal T-Atg7-/- mice (but not bone marrow chimeras as in Figure 3). While viral titers in the lungs of T-Atg7-/-mice are comparable to WT mice at day 3 of infection, they are significantly higher on day 6, suggesting T-Atg7-/- mice clear virus less efficiently than WT mice. This is probably due to the significantly lower absolute count of effector antigen-specific CD8 T cells. We expect that T-Atg7-/- mice would eventually clear the influenza virus as they survive the influenza challenge. The reason we did not measure this in a bone marrow chimera or adoptive transfer setting is because both the Atg7+/+ CD45.1 and Atg7-/- CDD45.2+ CD8 T cells would contribute toward viral clearance and it would be impossible to discern either's contribution. However, an additional mouse model as suggested by the reviewer, i.e. adoptive transfer experiments of Atg7-/- T cells into a T cell depleted host, would probably address this issue. This experiment could not be accomplished in the time frame given. We have now added the graph of viral clearance as Figure 1 to the manuscript.

*2) The authors use a model of heterotypic immunity against influenza to avoid a model in which antibody-mediated responses are involved in protection against viral challenge. However, it is notable that mouse survival is not affected by autophagy deficiency in T cells. The reviewers ask whether the authors have tried to use higher titers of the viruses or lethal strains such as those described before (Virology 2011, 412:36-45)*.

We too have thought about this experiment. However, even in wild type mice it is possible to overcome heterotypic immunity with a suitably intense secondary challenge. Furthermore, the dose we used is already at least 10,000 morbid doses and therefore, we think using a different strain also at a highly lethal dose is unlikely to give different results. Because of this, we feel that doing this experiment in knock out animals would not add any information on how these animals are protected.

*If antibodies are the main effectors against influenza and CD8 T cells are not sufficient or required for the protection against influenza challenge, this should be clarified by the authors*.

CD8 T cells are indeed required for the protection against the heterotypic influenza challenge as presented here in Figure 5. However, antibodies are essential for the protection against influenza infection such as Figures 2, 3 and 4. In fact a recent paper in Nature Medicine (ref 6) shows that mice with B cell–specific deletion of Atg7 (B/*Atg7*^-/-^ mice) showed normal primary antibody responses after immunization against influenza but failed to generate protective secondary antibody responses when challenged with influenza virus, resulting in high viral loads, widespread lung destruction and increased fatality. We have now made the requirement of CD8+ T cells clear in the Discussion.

In addition, the authors should more fully discuss the data on the role of human CD8 T cell responses in protection against influenza challenge after vaccines. The argument that the findings in mice may have relevance to diminished vaccine responses to flu vaccines in elderly relies on the relevance of CD8 T cell memory responses to vaccine efficacy. The final paragraph mentions briefly that CD8 T cell responses are an important correlate of vaccine protection; however, this should be discussed more extensively in light of the lack of in-depth experimental evidence in this study in the mouse for a role of memory CD8 T cells in protection against disease caused by viral challenge.

We fully agree with the reviewers and both these points (role of CD8 T cells in humans and the lack of depth knowledge about how mice are protected in this mouse model) have now been added to the Discussion, We fully agree with the reviewers and both these points (role of CD8 T cells in humans and the lack of depth knowledge about how mice are protected in this mouse model) have now been added to the Discussion.

*3) T/Atg7-/- mice show defects in T cell development and a reduction of CD4 and CD8 T cells in the spleen and lymph nodes (J. Immunol. 182:4046-4055 and*
Figure 2*). In*
Figure 1*, the authors should show the percentage and total numbers of antigen-specific CD8 T cells among all cells in the lung, not just among gated CD8 T cells. The authors conclude that there is “normal expansion of the antigen-specific effector CD8+ T cell compartment”. However, a general lymphopenia in CD4 and CD8 T cells in these mice should be clearly presented in*
Figure 1*, although a comparable fold of CD8 expansion was observed*.

We have now swapped Figures 1 and 2 to emphasize the lymphopenia in naïve mice at the start of the manuscript. In Figure 1 we clearly presented the lymphopenia both as proportions (in blood and LNs) and in absolute terms (in blood over time) for CD4+ and CD8+ T cells. We agree though that this was left out of Figure 2 where the antigen–specific response was described. So, we have now added absolute counts for influenza-challenged mice to Figure 2—figure supplement 1). The relevant reference from the Journal of Immunology has been added to the Results section.

*4) The reviewers feel that the interpretation of data in*
Figure 5
*is complicated by lymphopenia in T/Atg7-/- mice as well as autophagy deficiency in the CD4 T cell compartment. This point needs to be addressed in the authors' response*.

We agree with the reviewers that this remains a limitation of our study. Unfortunately, in the time frame, we could not generate BM chimera, vaccinate and challenge them. This point was therefore discussed as a limitation of our study at the end of the Results section of the description of Figure 5.

*5) In*
Figure 6*, the authors should consider immunodeficiency in aged CD4 T cells (Wakikawa et al., Exp Gerontol 34, 231-242; Cheung et al., Exp Gerontol 18, 451-460). Spermidine would target all cells types in vivo, including CD4+ T helper cells. The experiments performed do not distinguish between these possibilities*.

The reviewers are right in pointing out that spermidine also targets CD4+ T helper cells. In Figure 6 we only show the effect on antigen-specific CD8+ T cells as this is our focus in the KO model. In future experiments we plan to address if CD4 T helper cells are similarly affected by the loss of autophagy and can be modulated by spermidine treatment. We have added this to the Discussion.

*6) The knockout of essential genes from the autophagic cascade entails a major cellular dysfunction due to accumulating, dysfunctional mitochondria (and perhaps other organelles and protein aggregates). Hence, the phenotype of Atg7 knockout cells is not just the result of autophagy inhibition but rather that of accumulating damage. This problem should be discussed*.

We added a paragraph to the Discussion that deals with the problem of accumulation of damage as we, like the reviewer, think is a major part of why autophagy is important to keep the aged organism young.

*It would be even better to include some of the data that are mentioned as “not shown” with regard to the phenotype of mice lacking Atg5 in T cells*.

The phenotype as far as the naïve mouse is concerned has been verified using the more general vav promotor and another essential autophagy gene Atg5. As shown in the graphs below, Vav-Atg5-/- are both lymphopenic and accumulate the “virtual” memory compartment, which is identical to the phenotype observed in Vav-Atg7-/-. In fact all changes (including maturation of red blood cells, hematopoietic stem cells failure and myeloid bias, [31] and [55]) have been reproduced in this Atg5 mouse model, further confirming that changes are not Atg7 specific. We have added the most relevant data of Figure 1—figure supplement 2. Furthermore T cell lymphopenia in the absence of Atg5 has previously been shown by fetal liver chimera generated by Pua at al (JExp Med 2007), cited in the original manuscript.

*7) The authors only use one autophagy inducer, spermidine, in their experiments. Hence, it is still possible that autophagy would be only one of the spermidine effects (knowing that spermidine that might mediate additional metabolic effects on polyamine metabolism) that accounts for the improvement of CD8 memory T cells. The paper would be more complete, if the authors used other autophagy inducers (such as resveratrol, fasting etc.). At a minimum, the current limitation should be discussed*.

We tested no other autophagy inducers, although we appreciate the reviewer’s comments and the wise choice of alternative treatments. This study is merely a proof of concept that memory T cell responses can be improved by autophagy modulation. We would expect that either resveratrol and fasting would be able to achieve similar results and have added these ideas to the Discussion.

*8) Autophagic flux is only measured by the 'BDS feature', namely the co-localization of a lysosomal marker and LC3, measured by means of the Amnis Imagestream system. It is important to confirm these measurements (and in particular the efficiency of the ATG7 knockout) by immunoblot detection of the LC3 lipidation in the absence and presence of autophagy inhibitors*.

We have now performed the LC3 blot on purified CD8+ T cells from T-Atg7-/- and wild type mice and have added this blot to Figure 1—figure supplement 1 and d. It shows reduced LC3-II to LC3-I ratio in Atg7-/- T cells both in the absence and presence of the autophagy flux inhibitor Bafilomycin. We have also added a blot showing reduced expression of Atg7 in purified Atg7-/- CD8+ T cells.

*9) The controls in*
Figure 6
*look very different with regard to the proportion of LC3-I and LC3-II. This could be due to several different factors, including differences in basal autophagy conditions (density, nutritional factors) or sample processing (i.e. freeze-thawing samples will result in degradation of LC3-I which is more labile). The discrepancy in experimental conditions that results in such a notable discrepancy in the proportion of LC3-I and LC3-II in controls of the same cell type in different figures should be resolved*.

The controls in Figure 6 look different because they are different. All samples in Figure 6 but not in 6d have been treated with Bafilomycin A to see the reduction of the autophagic flux with DFMO more clearly. Below is a blot without BafA showing that basal levels are a little too low to clearly show a further reduction. The low concentration of 1µm of spd only induces further LC3 lipidation when cells are treated with DFMO. However, we feel that the original blots shown were better to bring the message across.Author response image 1.